# Evaluation of Zero-Valent Iron for Pb(II) Contaminated Soil Remediation: From the Analysis of Experimental Mechanism Hybird with Carbon Emission Assessment

**Junfang Sun [1], Angran Tian [2], Zheyuan Feng [2], Yu Zhang [3], Feiyang Jiang [4] and Qiang Tang [2,3,*]**

1  Dongwu Business School, Soochow University, Suzhou 215021, China; jfsun@suda.edu.cn
2  School of Rail Transportation, Soochow University, Suzhou 215131, China; gangguo@126.com (A.T.);
   fengzheyuan1999@sina.com (Z.F.)
3  Graduate School of Global Environmental Studies, Kyoto University, Kyoto 606-8501, Japan;
   zhangyu164@foxmail.com
4  D'Amore-McKim School of Business, Northeastern University, Boston, MA 02115, USA;
   jiang.fei@northeastern.edu
*  Correspondence: tangqiang@suda.edu.cn; Tel.: +86-18362676527

**Abstract:** Carbon emission is one of the main causes of global climate change, thus it is necessary to choose a low-carbon method in the contaminated soil remediation. This paper studies the adsorption ability of ZVI on Pb(II) contaminated soils under different working conditions. The removal efficiency of Pb(II) was 98% because of the suitable ZVI dosage, log reaction time and low initial solution concentration. The whole balancing process was much fast according to the pseudo-second-order kinetic and Freundlich isothermal model. Moreover, sequential extraction procedure (SEP) showed Pb(II) was transformed from Fe/Mn oxides-bound form to residual form in Pb(II) contaminated soils. From scanning electron microscopy (SEM), Brunauer-Emmett-Teller method (BET) and X-ray diffraction (XRD) results, it was confirmed that zero-valent iron (ZVI) stabilizes Pb(II) pollutants mostly through the combination of chemical adsorption and physical adsorption. The economic and carbon emission assessments were used to compare the cost and carbon emissions of different methods. The results show that ZVI adsorption has excellent economic benefits and low carbon emission.

**Keywords:** lead; zero-valent iron; adsorption; cost; carbon emission



## 1. Introduction

Increasingly serious global climate change is the huge challenge facing the world [1]. Carbon dioxide as the emission source of greenhouse gases has attracted extensive interest [2]. Previous studies have shown that the use of high embodied carbon building material is the main source of carbon emission in developing countries [3]. Therefore, reducing the use of high-carbon material is one of the main opportunities to reduce carbon emission and mitigate global climate change.

As a typical high-carbon material, cement has in common usage in contaminated soil remediation because of its low price and excellent effect [4]. By mixing it with contaminated soil, it can reduce the migration of pollutants to surrounding environment by solidification and stabilization [5,6]. However, considering its huge carbon emission, it is detrimental to the climate and needs to be replaced by a low-carbon, efficient and low-cost method.

Chemical precipitation, ion exchange, bioremediation and membrane separation techniques were commonly used to treat contaminated soil [7]. Among many treatment methods, the adsorption method has broad prospect due to its simple operation, high efficiency, low cost and low carbon [8]. There are currently a variety of adsorbent materials including biological, organic and inorganic adsorbent materials. Biological adsorbent materials have good adsorption capacity because they contain a large number of plant fibers, proteins and some

active functional groups such as -COOH, -OH. Organic materials reduce heavy metal pollution mainly by reducing water-soluble heavy metals content in soils and forming insoluble metal-organic complexes. The application of inorganic adsorbent materials in contaminated soil remediation is the most extensive due to its abundant reserves, low cost and good adsorption. At present, iron is the most widely used inorganic metal material and its consumption accounts for about 95% of all metal consumption. As a by-product of industry, zero-valent iron (ZVI) is extremely easy to obtain and has no secondary pollution to the environment. Thus, it is an ideal remediation material for contaminated soil and the climate [9].

Considering the wide range of lead (Pb) contaminated soil, Pb was used as a typical pollutant in this paper. Most of the iron powders used in the current research are nanoscale zero valent iron (NZVI) or industrial products, which are more expensive than industrial by-products [10]. Therefore, this paper selected ZVI to study its removal effects on Pb(II) in contaminated soils. The factors of dosage, initial concentration, reaction time and initial solution pH on the removal efficiency were taken into account. At the same time, the isothermal adsorption and adsorption kinetic process were studied. The removal mechanism of Pb(II) by ZVI was discussed in combination with SEP, SEM, BET and XRD results. The economic and carbon emission assessment was adopted to compare the cost and carbon emission from different methods.

## 2. Materials and Methods

### 2.1. Preparation of Materials and Characterization

The ZVI samples collected from a company in Suzhou were washed and dried. The dried samples were screened through 0.15 mm sieve. The ZVI sample is shown in Figure 1.

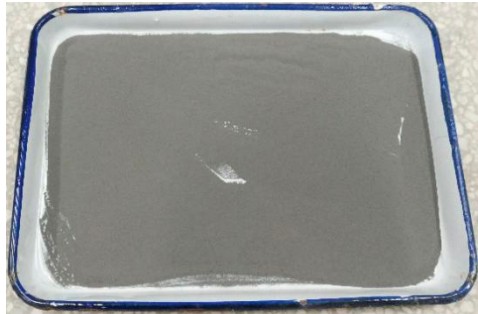

**Figure 1.** Appearance of the zero-valent iron (ZVI).

Soil samples were taken from Zhuzhou, Hunan. Figure 2a,b shows the geographical location, monthly average temperature and rainfall of the Zhuzhou. Soil samples were dried, crushed and sieved before taking experiments and its appearance is shown in Figure 2c. The reagent used in the experiments was Pb $(NO_3)_2$, it was dissolved in water to prepare standard solution and adjust the pH value with dilute nitric acid or sodium hydroxide. The diluted solutions were mixed with soil to form contaminated soil samples. The concentrations of Pb(II) were 3 mg/kg and 6 mg/kg, respectively.

The samples used in the experiment, including ZVI and Pb(II)-loaded ZVI, were scanned by XRD and their diffraction angles were in the range of 10–90 degrees. In addition, in order to test the microstructure, the ZVI and Pb(II)-loaded ZVI were observed by SEM. The specific surface area of ZVI and pH value of the solution were determined by BET test and the pH glass electrode, respectively.

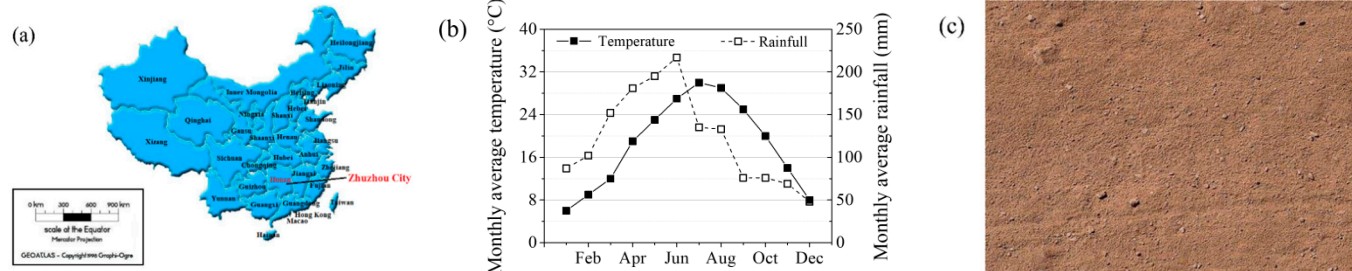

**Figure 2.** (**a**) Geographical location of Zhuzhou; (**b**) Monthly average temperature and rainfall of Zhuzhou; (**c**) Appearance of experimental soils.

### 2.2. Experimental Programs

### 2.2.1. Adsorption Experiments

The removal efficiency of ZVI for heavy metal contaminated soil was tested under different conditions, such as ZVI dosage, heavy metal concentration, reaction time and pH value. In order to reduce the experimental error, two groups of parallel experiments are set for each experiment and the average value was taken.

In the first set, different initial concentrations of $Pb(NO_3)_2$ solution were mixed with different dosages of ZVI. In the second set, two dosages of ZVI were respectively mixed with different concentrations of $Pb(NO_3)_2$ solution. In the third set, the ZVI was fixed and initial solution concentrations were ranged from 100 to 300 mg/L and the reaction time was changed. In the last set, the initial solution pH was changed. Except for the third set, the reaction time was 24 h.

All samples were agitated and centrifuged before the experiments. To measure the concentration, the supernatant was tested by Atomic Absorption Spectrophotometer (AAS). The solid residue was obtained by filtration and drying of the reaction solution (10 g/L ZVI dosage, 500 mg/L Pb(II) solution concentration) to carry out SEM, XRD and BET tests for mechanism study.

### 2.2.2. Incubation Experiments

In this paper, different amounts of ZVI were added into Pb(II) contaminated soil and cultured separately. 50 g Pb(II) contaminated soil was respectively weighed and placed into the 200 mL plastic bottle and then uniformly mixed with the ZVI powder. The addition amount of ZVI was maintained at 0% (blank sample) to 0.4% of the soil quality, respectively. At the same time, water was used to adjust the liquid-solid ratio to 50%. At $25 \pm 1$ °C and $95 \pm 1$% humidity, each sample was sealed and cured for 16 days [11]. The whole experiment was divided into 8 groups. After each sample was stable for 16 d, the Tessier five-step sequence extraction procedure was to measure the change of Pb(II) fraction in soil. The sequence extraction procedure had five steps, which divided heavy metals into five different forms [12]. Subsequently, the effect of ZVI availability in contaminated soils was further measured.

### 3. Results and Discussions

### 3.1. Adsorption Experimental Results

The relationship between the adsorption and the ZVI dosage is shown in Figure 3. With the increase of ZVI dose, the equilibrium concentration decreased gradually. The nonlinear relationship remained steady without obvious change after the dosage reached 10 g/L. When the concentration is reduced to a certain extent, the adsorption reaction will be difficult to proceed and increasing ZVI cannot effectively reduce the equilibrium concentration. The unit adsorption amount is the maximum when the ZVI dosage is 5 g/L. While the dosage is over 5 g/L, the adsorbent is in a relative excess state and the limiting factors were pollutants, it is difficult to promote the reaction by increasing ZVI dosage. Based on the trend of the variations of Qe and Ce, increasing ZVI dosage can promote the reaction but the overfull ZVI

leads to the inadequate utilization of the adsorbent. Considering the adsorption effect and material utilization efficiency, the optimal ZVI design dosage was 10 g/L.

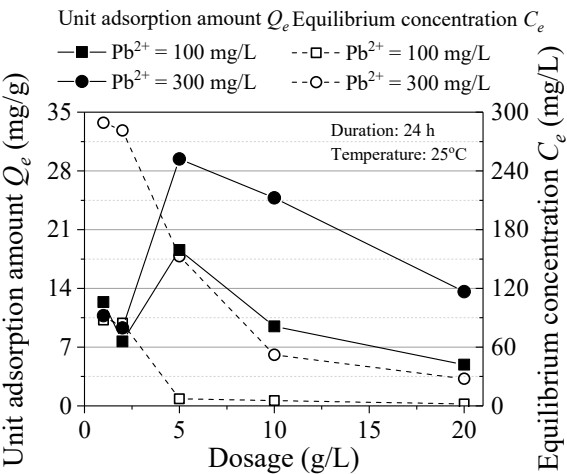

**Figure 3.** Effect of ZVI dosage.

The influence of the initial concentration is shown in Figure 4. The unit adsorption amount of ZVI increased with the initial concentration. Because of the high concentration, the collision probability between pollutants and ZVI is high and the active sites of ZVI are fully utilized. The maximum unit adsorption capacity was respectively 69.81 mg/g (ZVI = 10 g/L) and 45.46 mg/g (ZVI = 20 g/L). The adsorption efficiency relatively decreased with the increasing initial concentration, which was attributed to the mutual exclusion phenomenon of the ZVI surface. Increased concentration of contaminants on the surface of ZVI to increased repulsion between particles and the remaining adsorption sites are difficult to combine with contaminants. In addition, $H^+$ could react with ZVI and some hydroxides may precipitate on the surface of ZVI, reducing the reaction area [13]. Thence, it was feasible to form passivation film on the ZVI surface in high Pb(II) concentration environment, which was consistent with the study of Komnitsas [14].

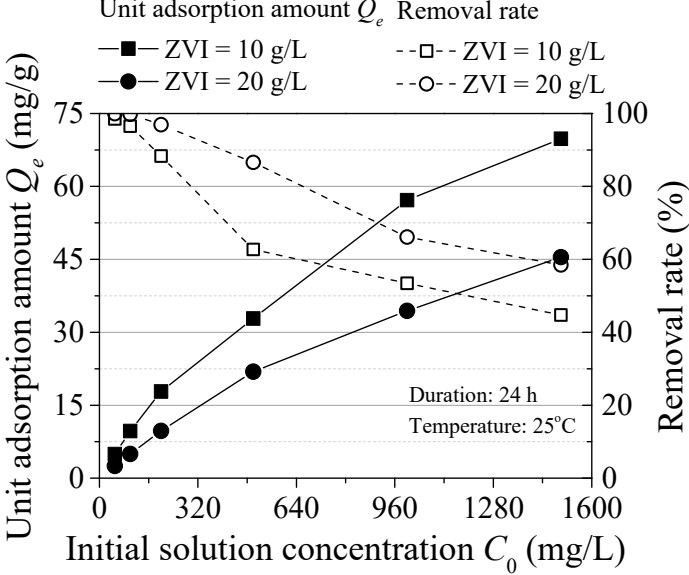

**Figure 4.** Effect of the initial concentration.

In order to further explore its adsorption capacity and adsorption mechanism, the data were fitted by Langmuir, Freundlich model and Redlich-Peterson model. Langmuir model

describes the comprehensive process of physical surface adsorption and chemical adsorption, which is below shown in Equation (1) [15]

$$Q_e = \frac{Q_m b C_e}{1 + b C_e},$$ (1)

where $Q_e$ is the equilibrium unit adsorption capacity, $Q_m$ is the maximum adsorption capacity and $b$ is the equilibrium constant.

Freundlich equation is used to consider the inhomogeneous surface interaction and the multi-molecular layer interaction, which is shown in Equation (2) [16,17]

$$Q_e = K C_e^{1/n},$$ (2)

where $K$ value is the adsorption capacity, $1/n$ represents the adsorption efficiency (between 0 to 1) and its value is the strength of the influence of concentration on adsorption capacity.

Considering the limitations of Langmuir and Freundlich isotherm models, empirical formulas of Redlich-Peterson (R-P) model is proposed. It consists of three parameters, which is shown in Equation (3)

$$Q_e = \frac{K_{RP} C_e}{1 + a_{RP} C_e^{\beta}},$$ (3)

where $K_{RP}$ and $a_{RP}$ represents the characteristic constant of Redlich-Peterson. B is an index between 0 and 1. When $\beta = 1$, it can be simplified as Langmuir model. When $\beta = 0$, it can be simplified as Henry model.

In Table 1, it is apparent that Freundlich model produced the best fit. The predicted adsorption capacity $K$ of ZVI is respectively 4.79 mg/g and 5.93 mg/g. Langmuir model indicates the maximum adsorption capacity $Q_m$ is up to 88.28 and 46.41 mg/g, respectively. The values of $K_{RP}$ and $a_{RP}$ predicted by R-P isotherm all indicate its excellent affinity for Pb(II).

**Table 1.** Isothermal model parameters for Pb(II) adsorption.

| Model | Freundlich Isothermal Model | | | Langmuir Isothermal Model | | | Redlich-Peterson Isothermal Model | | | |
|---|---|---|---|---|---|---|---|---|---|---|
| Dosage (g/L) | $K$ (mg/g) | $1/n$ | $R^2$ | $Q_m$ (mg/g) | $b$ (L/mg) | $R^2$ | $K_{RP}$ (L/g) | $a_{RP}$ (L/mg) | $\beta$ | $R^2$ |
| 10 | 4.79 | 0.40 | 0.99 | 88.28 | 0.0043 | 0.92 | 1.17 | 0.07 | 0.78 | 0.93 |
| 20 | 5.93 | 0.31 | 0.99 | 46.41 | 0.016 | 0.92 | 0.78 | 0.019 | 0.98 | 0.92 |

The effect of reaction time is shown in Figure 5. The Pb(II) adsorption process can be divided into fast stage (within 2 h) and slow reaction stage (from 2 to 10 h) and reaches to the equilibrium after 10 h. The maximum adsorption capacity reached 92.72% ($C_0$ = 100 mg/L) and 94.24% ($C_0$ = 300 mg/L) after the fast stage and the removal rate reached 98.17% and 92.16%, respectively. With the increase of initial concentration, the reaction equilibrium time changed little, while eventually the equilibrium concentration increased. The equilibrium adsorption time is shorter compared to some other inorganic materials. This may be because the effective adsorption area and sites are all occupied, which results in rapid adsorption.

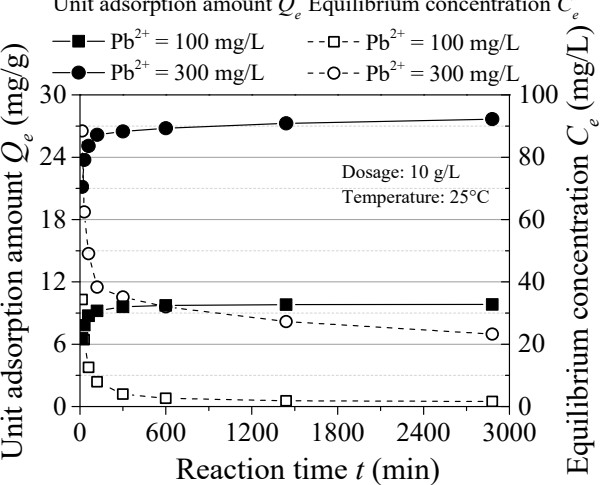

**Figure 5.** Effect of reaction time.

The adsorption kinetics equation describes the dynamic process of adsorption. The adsorption kinetic equations used in this paper include pseudo-first-order kinetic, pseudo-second-order kinetic and intraparticle diffusion model [18].

The pseudo-first-order kinetic equation is below shown in Equation (4):

$$Q_t = Q_e \left( 1 - e^{-k_1 t} \right), \tag{4}$$

where $k_1$ is the adsorption rate constant, $t$ is the reaction time, $Q_e$ and $Q_t$ are the adsorption amount at equilibrium and any time, respectively.

The pseudo-second-order kinetic equation is shown in Equation (5):

$$Q_t = \frac{k_2 Q_e^2 t}{1 + k_2 Q_e t}, \tag{5}$$

where $k_2$ is the adsorption rate constant.

The intraparticle diffusion model can accurately determine the control step of adsorbent adsorption rate. Its expression is shown in Equation (6):

$$Q_t = k_{int} t^{1/2} + C, \tag{6}$$

where $k_{int}$ represents the correlation adsorption rate constants and $C$ is intercept.

The predicted parameters are shown in Table 2. The equilibrium adsorption capacity (9.85 and 27.19 mg/g) is relatively higher than the pseudo-first-order model (9.50 and 26.47 mg/g). This model reflects the composite effect of multiple adsorption mechanisms and comprehensively represents the combination of the physical and chemical adsorption. The whole reaction is poorly fitted to the intragranular diffusion model, which indicates that it is not in the form of intragranular delivery.

**Table 2.** Kinetic model constants for Pb(II) adsorption.

| Model | Pseudo-First-Order Kinetic Model | | | Pseudo-Second-Order Kinetic Model | | | Intragranular Diffusion Model | | |
|---|---|---|---|---|---|---|---|---|---|
| Initial Concentration (mg/L) | $k_1$ (mg/g) | $Q_e$ (mg/g) | $R^2$ | $k_2$ (g/mg·min) | $Q_e$ (mg/g) | $R^2$ | $k_{int}$ (g/mg·min$^{1/2}$) | $C$ (mg/g) | $R^2$ |
| 100 | 0.069 | 9.50 | 0.86 | 0.013 | 9.85 | 0.99 | 0.047 | 7.98 | 0.41 |
| 300 | 0.099 | 26.47 | 0.78 | 0.008 | 27.19 | 0.98 | 0.093 | 23.68 | 0.50 |

Figure 6 shows the effect of initial solution pH on Pb(II) adsorption. When pH = 4 and 7, the removal rate is almost above 90%, which indicated the ZVI used in this experiment

can be widely applied. Besides, with the increase of pH, the removal rate of pollutants decreases relatively. This is because oxide or hydroxide is formed through the ZVI oxidation reaction and cover the surface of ZVI, which results in reducing the reaction contact area. Similarly, in the study of Zhang et al., when the pH was from 4 to 5, the removal rate of heavy metals was the highest, which was consistent with the experimental results [19].

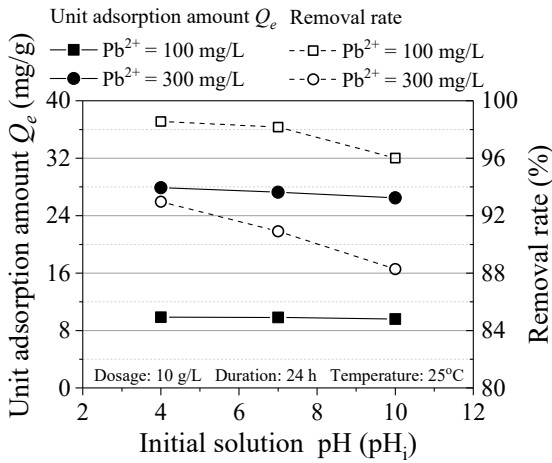

**Figure 6.** Effect of initial pH on the adsorption.

### 3.2. Incubation Experimental Results

In Figure 7, the main components of the contaminated soils were Fe/Mn oxides-bound fraction and residual fraction. In Figure 7a, the content of residual fraction and Fe/Mn oxides-bound fraction was respectively 18.78% and 81.21% without added ZVI. The content of residual fraction reached the lowest of 16.66% and Fe/Mn oxides-bound fraction reached the most of 83.33% when the ZVI dosage was 0.1%. As the ZVI dosage increased to 0.4%, the content of residual fraction became the largest at 31.3% and Fe/Mn oxides-bound fraction was reduced to minimum accordingly. In Figure 7b, the content of residual fraction and Fe/Mn oxides-bound fraction was respectively 4.08% and 95.91% without ZVI. When the dosage reached to 0.2%, the content of residual fraction and Fe/Mn oxides-bound fraction was respectively 3.5% and 96.49%. When it increased to 0.4%, the residual fraction was up to 3.85%. The leaching risk was the smallest according to the previous research [20,21]. As the amount of the ZVI increased, the content of residual fraction was generally on the rise. Based on the change trend, ZVI of 0.4% has better fixation effect on Pb(II) contaminated soil.

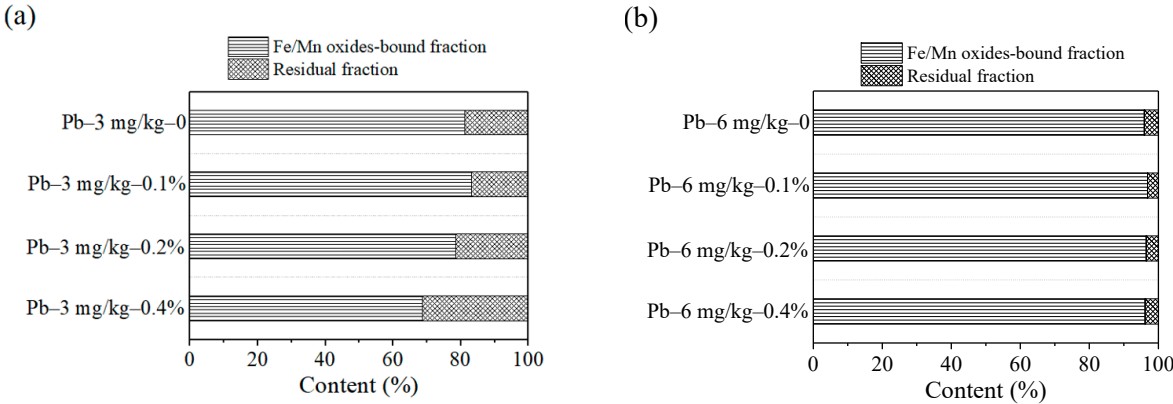

**Figure 7.** Dosage effect on Pb speciation: (**a**) 3 mg/kg; (**b**) 6 mg/kg.

### 3.3. Mechanism Discussions

Figure 8 shows the SEM images of the ZVI. It is obvious that the ZVI particles were distributed in the block form. After adsorption, a layer of small particulate matter adhered to the ZVI surface. This is because the intergranular structure changes and flocculation or crystal precipitation is formed by surface complexation or precipitation mechanism [22].

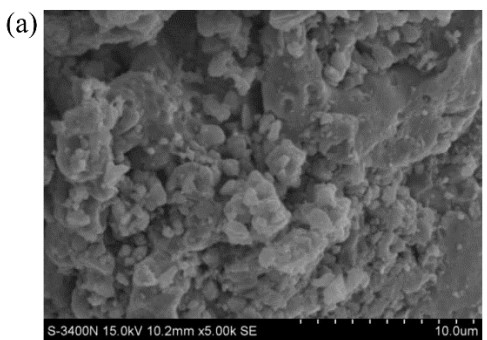
(a)

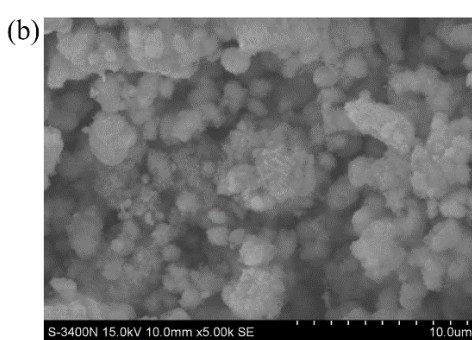
(b)

**Figure 8.** SEM images: (**a**) Before adsorption; (**b**) After adsorption.

The BET results are shown in Figure 9. The SSA was 152.6 $m^2/g$ by $N_2$ adsorption isotherm. ZVI with a large SSA is beneficial to Pb(II) adsorption due to the sufficient contact between adsorbent and adsorbate. The pore size determines the amount of adsorption and the mesoporous structure can promote the physical adsorption and increase the adsorption amount of pollutants. The pore size was mainly distributed in 10 to 15 nm (content was 54%), where the maximum pore volume was up to 0.0049 $cm^3/g$. This indicated Pb(II) can diffuse faster in the mesopores of ZVI, which results in increasing the adsorption capacity of ZVI for Pb(II).

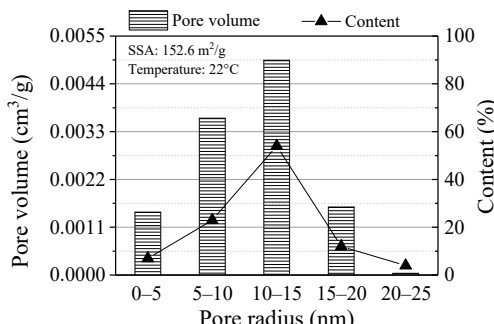

**Figure 9.** BET results of the ZVI.

Figure 10 shows the XRD pattern of the ZVI before and after reaction. The peak diffraction intensity appears at 44.6°, 65.0° and 82.5°, which indicated the presence of ZVI. According to the semi quantitative analysis, the amount of ZVI before reaction was more than 99%. After reaction, characteristic peaks of ZVI were still and its diffraction intensity was greatly reduced, which indicated that ZVI was consumed in adsorption. Besides, new characteristic peaks appear at the different diffraction angles, which indicated that new substances could be generated. Among the newly formed substances, the most content of FeO(OH) component is 76.1% and its diffraction angles was respectively 16.4°, 27.3°, 34.3° and 60.1°. FeO(OH) can be decomposed into $Fe_2O_3$, which was consistent with the study of He [23]. At the same time, the diffraction peaks of $\alpha$-$Fe_2O_3$ appear at the $2\theta = 33.5°$ and 50°. Zhou also showed that after ZVI treatment of heavy metals contaminated groundwater, the composition of $Fe_3O_4$-$Fe_2O_3$ mixture increased [24].

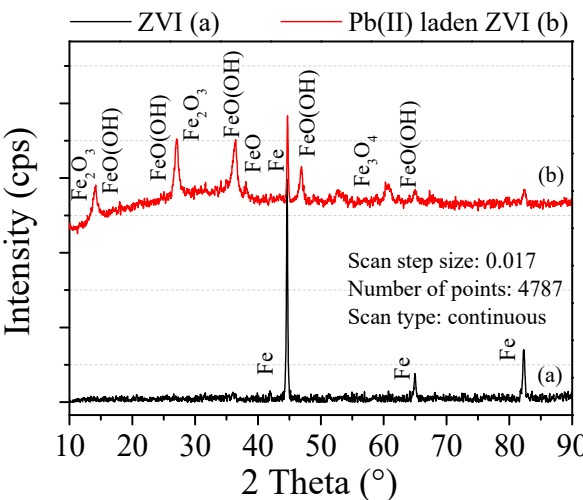

**Figure 10.** XRD pattern of the ZVI.

Under the standard conditions of 298 K and 101 kPa, the standard electrode potential of Fe and Pb(II) is $-0.440$ V and $-0.126$ V, respectively. ZVI is oxidized ($Fe^0 \rightarrow Fe^{2+} + 2e^-$) and Pb(II) is fixed by transferring electrons to each other on the ZVI surface. However, the standard electrode potentials of Pb(II) and Fe are not much different and the fixation of Pb(II) is also affected through other mechanisms.

When Fe is oxidized in Pb(II) solution, $FeOH^+$ may be formed and then some precipitations such as $Pb(OH)_2$, $PbO_2$ and $PbO$ can be formed by complexing with Pb(II) in the solution. The reaction formula is written as

$$Fe^{2+} + H_2O \rightarrow FeOH^+ + H^+ \tag{7}$$

$$Pb^{2+} + FeOH^+ \rightarrow PbOH^+ + Fe^{2+} \tag{8}$$

$$PbOH^+ + H_2O \rightarrow Pb(OH)_2(s) + H^+. \tag{9}$$

At the same time, Fe(II) is easily oxidized to Fe(III) in solution and then combines with $OH^+$ to form $\alpha$-FeOOH and $\beta$-FeOOH complexes. The newly formed complexes have strong flocculation and coprecipitation with Pb(II). It should be noted that no Pb(II) element or compound was detected in the XRD spectra. This is because the Pb(II)-containing substance was set to a low content under experimental conditions and existed in a dispersed form.

Based on all above results, part ZVI participates in the physical adsorption and does not change its existing form. Pb(II) is immobilized on the ZVI surface by surface bonding and electrostatic interaction. According to the difference of reaction speed in the two stages of adsorption kinetics, it is known that physical adsorption was not a single process. Another part of ZVI is involved in the chemical adsorption and is mainly based on chemisorption. ZVI is consumed by the reaction and some new substances such as complexes and precipitations are produced.

## 4. Economical and Carbon Emission Evaluation

### 4.1. Economical Evaluation

Table 3 shows the efficiency and cost of the different remediation methods based on the extensive investigation and economic analysis methods [25–27]. Compared to other methods, it is obvious that chemical remediation takes less time in spite of higher cost. However, considering the ZVI was regarded as a by-product of industry, it can be easily obtained at extremely low cost. As a result, the utilization of ZVI provides the potential to further reduction of the chemical remediation cost.

**Table 3.** The efficiency and cost of the different remediation methods.

| | Remediation Methods | Remediation Cycle | Cost (U.S.) | Cost (China) |
|---|---|---|---|---|
| Chemical remediation | Ex-Situ Solidification/Stabilization | 100–1200 m$^3$/day | 90–245 US dollars/m$^3$ | 500–1500 RMB/m$^3$ |
| | Ex-Situ Chemical Oxidization/Reduction | several weeks or months | 200–660 US dollars/m$^3$ | 500–1500 RMB/m$^3$ |
| | Ex-Situ Soil Washing | 3–12 months | 53–420 US dollars/m$^3$ | 1000–1500 RMB/m$^3$ |
| | In–Situ Solidification/Stabilization | 3–6 months | 50–330 US dollars/m$^3$ | 600–1000 RMB/m$^3$ |
| | In-Situ Chemical Oxidation & Reduction | 3–24 months | 120–200 US dollars/m$^3$ | 500–2000 RMB/m$^3$ |
| Bioremediation | Soil Phytoremediation | 3–8 years | 25–100 US dollars/t | 100–400 RMB/t |
| Physical remediation | Ex-Situ Solidification/Stabilization | several weeks or years | 50–300 US dollars/m$^3$ | 600–2000 RMB/t |

Data source: The USEPA.

### 4.2. Carbon Emission Assessment

The whole process of soil remediation includes four stages: material production, transportation, construction and disposal. The total carbon emission is the sum of the four stages as shown in Equations (10)–(13) [28–31]:

$$CE(S) = CE(S_1) + CE(S_2) + CE(S_3) + CE(S_4) \tag{10}$$

$$CE(S_1) = \sum_i (1 + \varphi_{1i}) \times Q_{Mi} \times C_{E1i} \tag{11}$$

$$CE(S_2) = \sum_i (1 + \varphi_{2i}) \times Q_{Ti} \times C_{E2i} \tag{12}$$

$$CE(S_3) = \sum_i Q_{Pj} \times C_{E3j} \tag{13}$$

$$CE(S_4) = \sum_i Q_{si} \times C_{E4i}, \tag{14}$$

where $CE(S)$, $CE(S_1)$, $CE(S_2)$, $CE(S_3)$ and $CE(S_4)$ are the carbon emission at the whole process, material production, transportation, construction and disposal, $\varphi_{1i}$ and $\varphi_{2i}$ are the percentage of wastes at material production and transportation, $Q_{Mi}$, $Q_{Ti}$, $Q_{Pj}$ and $Q_{Si}$ are the net quantity of material at material production, transportation, construction and disposal, $C_{E1i}$, $C_{E2i}$, $C_{E3j}$ and $C_{E4i}$ are the emission factors at material production, transportation, construction and disposal and $i$ and $j$ is the type of materials and energy, respectively.

Considering that the carbon emissions from bioremediation and chemical oxygenation/reduction can be ignored, only the carbon emissions from solidification/stabilization, thermal desorption and ZVI adsorption were calculated. The emission factors are shown in Table 4 [32–35].

**Table 4.** Emission factors.

| | Unit | CO$_2$ Emission Factors (kg/unit) |
|---|---|---|
| Gasoline | kg | 2.907 |
| Diesel | kg | 3.097 |
| Cement | kg | 0.73 |
| Electric Energy | kwh | 0.59 |

It is assumed that the depth of contaminated soil is 30 cm and the transportation distance of remote remediation is 30 km, the cement content in Solidification/stabilization is 5%, the power consumption of thermal destruction is 1000 kwh/m$^3$ and ZVI adsorption is ploughed by tractor for three times. The calculation results are shown in Table 5.

**Table 5.** $CO_2$ emission of the different remediation methods.

| Remediation Methods | $CO_2$ Emission (kg/1000 m$^2$) |
|---|---|
| Ex-Situ Solidification/Stabilization | 98,266.10 |
| In-Situ Solidification/Stabilization | 97,121.94 |
| Ex-Situ Thermal Desorption | 178,009.96 |
| In-Situ ZVI Adsorption | 100.40 |

By comparing different remediation methods, the carbon emission of ZVI adsorption is very small compared with other methods. When the contaminated soil area is 1000 m$^2$, it is about 0.10% of the consolidation/stabilization and 0.06% of the thermal degradation. Moreover, the advantages become more obvious with the increase of the area.

**5. Conclusions**

In this paper, under normal temperature and atmospheric pressure, ZVI was adopted to treat the Pb(II) contaminated soil. The effects of ZVI dosage, soil contamination degree, treatment time and initial pH on the removal efficiency were obtained. The isothermal adsorption process and kinetic properties were studied. The removal mechanism by ZVI was discussed.

The removal rate increased with ZVI dosage, prolongation of reaction time and decrease of contamination degree but was little affected by solution pH. The removal rate reaches more than 98%, the predicted unit adsorption capacity can reach 88.28 mg/g. The reaction process proceeds rapidly, which is determined by pseudo-second-order kinetic and Freundlich model. SEM, BET and XRD revealed the adsorption mechanism of ZVI on Pb(II), including reduction of Pb(II) by ZVI, coprecipitation of iron-containing colloid with Pb(II) and electrostatic interaction.

In addition, economic and carbon emission evaluations were used to compare the costs and carbon emissions of different methods. The results show that the price of ZVI adsorption is very low considering that it is an industrial by-product. Compared with solidification/stabilization and thermal destruction, when the area is 1000 m$^2$, its carbon emission is less than 0.1% of their own. With the increase of the area, its advantages become more obvious. Therefore, ZVI adsorption has an excellent application prospect.

For further study, the adsorption capacity of ZVI for other kinds of heavy metals and organic pollutants needs to be evaluated. In order to verify the remediation effect of ZVI to contaminated soil in a large scale, field tests need to be carried out.

**Author Contributions:** Conceptualization: J.S. and Q.T.; methodology: J.S., A.T. and Z.F.; validation, Y.Z. and Z.F.; formal analysis: J.S., A.T. and F.J.; investigation: Y.Z. and F.J.; resources: Z.F. and Y.Z.; data curation: Y.Z., A.T. and F.J.; writing—original draft preparation: J.S., A.T., Y.Z. and F.J.; writing—review and editing: J.S., A.T., Y.Z., Z.F. and F.J.; supervision: Q.T. and Y.Z.; project administration: J.S. and Q.T. All authors have read and agreed to the published version of the manuscript.

**Funding:** The research presented herein is supported by the National Nature Science Foundation of China (52078317), Natural Science Foundation of Jiangsu Province (BK20170339), project from Jiangsu Provincial Department of Housing and Urban-Rural Development (2020ZD05) and Bureau of Housing and Urban-Rural Development of Suzhou (2020-15).

**Institutional Review Board Statement:** Not applicable.

**Informed Consent Statement:** Not applicable.

**Data Availability Statement:** The research data supporting this publication are given within this paper and as supplementary material.

**Conflicts of Interest:** The authors declare no conflict of interest.

## Abbreviations

| | |
|---|---|
| $C_e$ | equilibrium concentration (mg/L) |
| $C_0$ | initial solution concentration (mg/L) |
| $t$ | rection time (min) |
| $Q_e$ | unit adsorption capacity (mg/g) |
| $Q_m$ | maximum adsorption capacity (mg/g) |
| $b$ | equilibrium constant (L/mg) |
| $K$ | adsorption capacity value (mg/g) |
| $1/n$ | adsorption efficiency |
| $K_{RP}$ | characteristic constant of Redlich-Peterson (L/g) |
| $a_{RP}$ | characteristic constant of Redlich-Peterson (L/mg) |
| $\beta$ | adsorption index |
| $k_1$ | adsorption rate constant of pseudo-first-order model (mg/g) |
| $k_2$ | adsorption rate constant (g/mg·min) |
| $k_{int}$ | correlation adsorption rate constant of pseudo-second-order model (g/mg·min$^{1/2}$) |
| $C$ | intercept (mg/g) |
| $CE(S)$ | carbon emission at the whole process (kg) |
| $CE(S_1)$ | carbon emission at material production (kg) |
| $CE(S_2)$ | carbon emission at transportation (kg) |
| $CE(S_3)$ | carbon emission at construction (kg) |
| $CE(S_4)$ | carbon emission at disposal (kg) |
| $\varphi_{1i}$ | percentage of wastes at material production (%) |
| $\varphi_{2i}$ | percentage of wastes at transportation (%) |
| $Q_{Mi}$ | net quantity of material at material production (unit) |
| $Q_{Ti}$ | net quantity of material at transportation (unit) |
| $Q_{Pi}$ | net quantity of material at construction (unit) |
| $Q_{Si}$ | net quantity of material at disposal (unit) |
| $C_{E1i}$ | emission factors at material production (kg/unit) |
| $C_{E2i}$ | emission factors at transportation (kg/unit) |
| $C_{E3i}$ | emission factors at construction (kg/unit) |
| $C_{E4i}$ | emission factors at disposal (kg/unit) |

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
