# Peer review of "Evaluation of Zero-Valent Iron for Pb(II) Contaminated Soil Remediation: From the Analysis of Experimental Mechanism Hybird with Carbon Emission Assessment"

_sustainability, doi:10.3390/su13020452_

Round 1
Reviewer 1 Report
The article: "Evaluation of Zero-valent Iron for Pb (II) Contaminated Soil Remediation: From the Analysis of Experimental Mechanism Hybird with Carbon Emission Assessment" is interesting and well structured but needs a revision.
• For reasons of clarity, a nomenclature should be included containing all the symbols used in this manuscript;
• In the conclusions section, it is recommended to briefly discuss the future research that the authors intend to address.
Author Response
From: School of Rail Transportation, Soochow University, China
Dec. 30, 2020
To: Sustainability
Dear Editor,
We would like to thank the editorial board of Sustainability for the comments on the manuscript titled " Evaluation of Zero-valent Iron for Pb(II) Contaminated Soil Remediation: From the Analysis of Experimental Mechanism Hybird with Carbon Emission Assessment ".
We thank the reviewers for their careful reading and helpful comments on previous draft. We have carefully taken their comments into consideration in preparing our revision, which has resulted in a paper that is clearer, more compelling and broader.
As following, you can find our responses to the reviewers’ comments, and the modifications in revision were also highlighted. Thanks very much for your support.
Best wishes
Sincerely yours,
TANG Qiang
Ph.D., Professor, Department Head
Civil and Environmental Engineering, School of Rail Transportation, Soochow University
No.8 Jixue Road, Xiangcheng District, Suzhou, 215131, China
Email: tangqiang@suda.edu.cn
Responses to reviewers’ comments
Reviewer’s comments:
1. For reasons of clarity, a nomenclature should be included containing all the symbols used in this manuscript. |
Answer:
Thanks very much for editor's comments. The nomenclature is able to improve the clarity of the paper, and it was supplemented at the end of the manuscript based on the comment. The relevant revisions were highlighted in the manuscript.
2. In the conclusions section, it is recommended to briefly discuss the future research that the authors intend to address. |
Answer:
Thanks very much for editor's comment. The future research discussion plays an important role in improving the structure of the paper. The reviewers’ comments were fully accepted, and the future research was discussed and highlighted in the conclusion section, as shown follows:
“For further study, the adsorption capacity of ZVI for other kinds of heavy metals and organic pollutants needs to be evaluated. In order to verify the remediation effect of ZVI to contaminated soil in a large scale, field tests need to be carried out.”
Reviewer’s comments:
1. First of all, I recommend for the authors to check doi:10.3390/ijerph17165817 and then highlight the novelty of the tests conducted. |
Answer:
Thanks very much for reviewer's comment. In previous studies, nanoscale zero valent iron (NZVI) or industrial products was often used to treat organic pollutants. Different from the traditional way, this paper uses industrial by-product iron powder to remediate heavy metal contaminated soil, which can reduce the cost. This paper was cited and the related discussion was added in the revised manuscript, as shown below:
“Most of the iron powders used in the current research are nanoscale zero valent iron (NZVI) or industrial products, which are more expensive than industrial by-products[10].”
2. Secondly, I would like to question the methodology about adsorption: what was the time of the experiments? Was the time kept qual for all tested cases as it is shown at fig. 5? |
Answer:
Thanks very much for reviewer's comment. The experiments were carried out in July 2020. Except for the third set, the reaction time was 24 hours, which was shown in fig 1, 2, 3, 4 and 6. The reaction time was supplemented and highlighted in experimental programs section, as shown follows:
“Except for the third set, the reaction time was 24 hours.”
3. Please correct fig. 6. |
Answer:
Thanks very much for reviewer's comment. The fig. 6 was corrected based on the comment. The related discussion was modified and highlighted in the revised manuscript, as shown below:
“When pH = 4 and 7, the removal rate is almost above 90%, which indicated the ZVI used in this experiment can be widely applied.”
4. Please simplify tab. 5. There is no need to give the results of such simple calculations. One example is enough. |
Answer:
Thanks very much for reviewer's comment. The reviewers’ comments were fully accepted. Tab. 5 was simplified and only one result was retained.

Reviewer 2 Report
The manuscript entitled “Evaluation of Zero-valent Iron for Pb(II) Contaminated Soil Remediation: From the Analysis of Experimental Mechanism Hybird with Carbon Emission Assessment” presents an interesting but already known method of soil remediation by ZVI.
First of all, I recommend for the authors to check doi:10.3390/ijerph17165817 and then highlight the novelty of the tests conducted.
Secondly, I would like to question the methodology about adsorption: what was the time of the experiments? Was the time kept qual for all tested cases as it is shown at fig. 5?
Please correct fig. 6.
Please simplify tab. 5. There is no need to give the results of such simple calculations. One example is enough.
Author Response

(The authors gave the same response as above.)
